# Neural Contextual Bandits with Deep Representation and Shallow Exploration

## Abstract

We study neural contextual bandits, a general class of contextual bandits, where each context-action pair is associated with a raw feature vector, but the specific reward generating function is unknown. We propose a novel learning algorithm that transforms the raw feature vector using the last hidden layer of a deep ReLU neural network (deep representation learning), and uses an upper confidence bound (UCB) approach to explore in the last linear layer (shallow exploration). We prove that under standard assumptions, our proposed algorithm achieves $\widetilde{O}(\sqrt{T})$ finite-time regret, where $T$ is the learning time horizon. Compared with existing neural contextual bandit algorithms, our approach is computationally much more efficient since it only needs to explore in the last layer of the deep neural network.

## 1 Introduction

Multi-armed bandits (MAB) [9, 8, 30] are a class of online decision-making problems where an agent needs to learn to maximize its expected cumulative reward while repeatedly interacting with a partially known environment. Based on a bandit algorithm (also called a strategy or policy), in each round, the agent adaptively chooses an arm, and then observes and receives a reward associated with that arm. Since only the reward of the chosen arm will be observed (bandit information feedback), a good bandit algorithm has to deal with the exploration-exploitation dilemma: trade-off between pulling the best arm based on existing knowledge/history data (exploitation) and trying the arms that have not been fully explored (exploration).

In many real-world applications, the agent will also be able to access detailed contexts associated with the arms. For example, when a company wants to choose an advertisement to present to a user, the recommendation will be much more accurate if the company takes into consideration the contents, specifications, and other features of the advertisements in the arm set as well as the profile of the user. To encode the contextual information, contextual bandit models and algorithms have been developed, and widely studied both in theory and in practice [19, 39, 34, 16, 1]. Most existing contextual bandit algorithms assume that the expected reward of an arm at a context is a linear function in a known context-action feature vector, which leads to many useful algorithms such as LinUCB [16], OFUL [1], etc. The representation power of the linear model can be limited in applications such as marketing, social networking, clinical studies, etc., where the rewards are usually counts or binary variables. The linear contextual bandit problem has also been extended to richer classes of parametric bandits such as the generalized linear bandits [24, 35] and kernelised bandits [44, 15].

With the prevalence of deep neural networks (DNNs) and their phenomenal performances in many machine learning tasks [32, 25], there has emerged a line of work that employs DNNs to increase the representation power of contextual bandit algorithms [5, 38, 17, 49, 52, 20, 51]. The problems they solve are usually referred to as *neural contextual bandits*. For example, Zhou et al. [52] developed the NeuralUCB algorithm, which can be viewed as a natural extension of LinUCB [16, 1], where they

use the output of a deep neural network with the feature vector as input to approximate the reward. Zhang et al. [51] adapted neural networks in Thompson Sampling [43, 14, 40] for both exploration and exploitation and proposed NeuralTS . For a fixed time horizon $T$, it has been proved that both NeuralUCB and NeuralTS achieve a $O(\widetilde{d}\sqrt{T})$ regret bound, where $\widetilde{d}$ is the effective dimension of a neural tangent kernel matrix which can potentially scale with $O(TK)$ for $K$-armed bandits. This high complexity is mainly due to that the exploration is performed over the entire huge neural network parameter space, which is inefficient and even infeasible when the number of neurons is large. A more realistic and efficient way of learning neural contextual bandits may be to just explore different arms using the last layer as the exploration parameter. More specifically, Riquelme et al. [38] provided an extensive empirical study of benchmark algorithms for contextual-bandits through the lens of Thompson Sampling, which suggests decoupling representation learning and uncertainty estimation improves performance.

In this paper, we show that the decoupling of representation learning and the exploration can be theoretically validated. We study a new neural contextual bandit algorithm, which learns a mapping to transform the raw features associated with each context-action pair using a deep neural network (*deep representation*), and then performs an upper confidence bound (UCB)-type exploration over the linear output layer of the network (*shallow exploration*). We prove a sublinear regret of the proposed algorithm by exploiting the UCB exploration techniques in linear contextual bandits [1] and the analysis of deep overparameterized neural networks using neural tangent kernels [27]. Our theory confirms the empirically observed effectiveness of decoupling the deep representation learning and the UCB exploration in contextual bandits [38, 49].

**Contributions** we summarize the main contributions of this paper as follows.

- We propose a contextual bandit algorithm, Neural-LinUCB, for solving a general class of contextual bandit problems without knowing the specific reward generating function. The proposed algorithm learns a deep representation to transform the raw feature vectors and performs UCB-type exploration in the last layer of the neural network, which we refer to as deep representation and shallow exploration. Compared with LinUCB [34, 16] and neural bandits such as NeuralUCB [52] and NeuralTS [51], our algorithm enjoys the best of two worlds: strong expressiveness due to the deep representation and computational efficiency due to the shallow exploration.

- Despite the usage of a DNN as the feature mapping, we prove a $\widetilde{O}(\sqrt{T})$ regret for the proposed Neural-LinUCB algorithm, which matches the regret bound of linear contextual bandits [16, 1]. To the best of our knowledge, this is the first work that theoretically shows the convergence of bandits algorithms under the scheme of deep representation and shallow exploration. It is notable that a similar scheme called Neural-Linear was proposed by Riquelme et al. [38] for Thompson sampling algorithms, and they empirically showed that decoupling representation learning and uncertainty estimation improves the performance. Our work confirms this observation from a theoretical perspective.

- We conduct experiments on contextual bandit problems based on real-world datasets, demonstrating a better performance and computational efficiency of Neural-LinUCB over LinUCB and NeuralUCB, which well aligns with our theory.

## 1.1 Additional related work

There is a line of related work to ours on the recent advance in the optimization and generalization analysis of deep neural networks. In particular, Jacot et al. [27] first introduced the neural tangent kernel (NTK) to characterize the training dynamics of network outputs in the infinite width limit. From the notion of NTK, a fruitful line of research emerged and showed that loss functions of deep neural networks trained by (stochastic) gradient descent can converge to the global minimum [22, 4, 21, 54, 53]. The generalization bounds for overparameterized deep neural networks are also established in Arora et al. [6, 7], Allen-Zhu et al. [3], Cao and Gu [12, 13]. Recently, the NTK based analysis is also extended to the study of sequential decision problems including bandits [52, 51], and reinforcement learning algorithms [11, 36, 45, 47].

Our algorithm is also different from Langford and Zhang [29], Agarwal et al. [2] which reduce the bandit problem to supervised learning. Moreover, their algorithms need to access an oracle that returns the optimal policy in a policy class given a sequence of context and reward vectors, whose regret depends on the VC-dimension of the policy class.

**Notation** We use $[k]$ to denote a set $\{1, \ldots, k\}$, $k \in \mathbb{N}^+$. $\|\mathbf{x}\|_2 = \sqrt{\mathbf{x}^\top \mathbf{x}}$ is the Euclidean norm of a vector $\mathbf{x} \in \mathbb{R}^d$. For a matrix $\mathbf{W} \in \mathbb{R}^{m \times n}$, we denote by $\|\mathbf{W}\|_2$ and $\|\mathbf{W}\|_F$ its operator norm and Frobenius norm respectively. For a semi-definite matrix $\mathbf{A} \in \mathbb{R}^{d \times d}$ and a vector $\mathbf{x} \in \mathbb{R}^d$, we denote the Mahalanobis norm as $\|\mathbf{x}\|_\mathbf{A} = \sqrt{\mathbf{x}^\top \mathbf{A} \mathbf{x}}$. Throughout this paper, we reserve the notations $\{C_i\}_{i=0,1,\ldots}$ to represent absolute positive constants that are independent of problem parameters such as dimension, sample size, iteration number, step size, network length and so on. The specific values of $\{C_i\}_{i=0,1,\ldots}$ can be different in different context. For a parameter of interest $T$ and a function $f(T)$, we use notations such as $O(f(T))$ and $\Omega(f(T))$ to hide constant factors and $\widetilde{O}(f(T))$ to hide constant and logarithmic dependence of $T$.

## 2 Preliminaries

In this section, we provide the background of contextual bandits and deep neural networks.

### 2.1 Linear contextual bandits

A contextual bandit is characterized by a tuple $(\mathcal{S}, \mathcal{A}, r)$, where $\mathcal{S}$ is the context (state) space, $\mathcal{A}$ is the arm (action) space, and $r$ encodes the unknown *reward generating function* at all context-arm pairs. A learning agent, who knows $\mathcal{S}$ and $\mathcal{A}$ but does not know the true reward $r$ (values bounded in $(0, 1)$ for simplicity), needs to interact with the contextual bandit for $T$ rounds. At each round $t = 1, \ldots, T$, the agent first observes a context $s_t \in \mathcal{S}$ chosen by the environment; then it needs to adaptively select an arm $a_t \in \mathcal{A}$ based on its past observations; finally it receives a reward $\widehat{r}_t(\mathbf{x}_{s,a_t}) = r(\mathbf{x}_{s,a_t}) + \xi_t$, where $\mathbf{x}_{s,a} \in \mathbb{R}^d$ is a known feature vector for context-arm pair $(s, a) \in \mathcal{S} \times \mathcal{A}$, and $\xi_t$ is a random noise with zero mean. The agent's objective is to maximize its expected total reward over these $T$ rounds, which is equivalent to minimizing the pseudo regret [8]:

$$R_T = \mathbb{E}\left[ \sum_{t=1}^T \left( \widehat{r}(\mathbf{x}_{s_t, a_t^*}) - \widehat{r}(\mathbf{x}_{s_t, a_t}) \right) \right], \tag{2.1}$$

where $a_t^* \in \mathrm{argmax}_{a \in \mathcal{A}}\{r(\mathbf{x}_{s_t,a}) = \mathbb{E}[\widehat{r}(\mathbf{x}_{s_t,a})]\}$. To simplify the exposition, we use $\mathbf{x}_{t,a}$ to denote $\mathbf{x}_{s_t,a}$ since it only depends on the round index $t$ in most bandit problems, and we assume $\mathcal{A} = [K]$.

In some practical problems, the agent has a prior knowledge that the reward-generating function $r$ has some specific parametric form. For instance, in linear contextual bandits, the agent knows that $r(\mathbf{x}_{s,a}) = \mathbf{x}_{s,a}^\top \boldsymbol{\theta}^*$ for some unknown weight vector $\boldsymbol{\theta}^* \in \mathbb{R}^d$. One provably sample efficient algorithm for linear contextual bandits is Linear Upper Confidence Bound (LinUCB) [1]. Specifically, at each round $t$, LinUCB chooses action by the following strategy

$$a_t = \underset{a \in [K]}{\mathrm{argmax}} \left\{ \mathbf{x}_{t,a}^\top \boldsymbol{\theta}_t + \alpha_t \|\mathbf{x}_{t,a}\|_{\mathbf{A}_t^{-1}} \right\},$$

where $\boldsymbol{\theta}_t$ is a point estimate of $\boldsymbol{\theta}^*$, $\mathbf{A}_t = \lambda \mathbf{I} + \sum_{i=1}^t \mathbf{x}_{i,a_i} \mathbf{x}_{i,a_i}^\top$ with some $\lambda > 0$ is a matrix defined based on the historical context-arm pairs, and $\alpha_t > 0$ is a tuning parameter that controls the exploration rate in LinUCB.

### 2.2 Deep neural networks

In this paper, we use $f(\mathbf{x})$ to denote a neural network with input data $\mathbf{x} \in \mathbb{R}^d$. Let $L$ be the number of hidden layers and $\mathbf{W}_l \in \mathbb{R}^{m_l \times m_{l-1}}$ be the weight matrices in the $l$-th layer, where $l = 1, \ldots, L$, $m_1 = \ldots = m_{L-1} = m$ and $m_0 = m_L = d$. Then a $L$-hidden layer neural network is defined as

$$f(\mathbf{x}) = \sqrt{m} \boldsymbol{\theta}^{*\top} \sigma_L(\mathbf{W}_L \sigma_{L-1}(\mathbf{W}_{L-1} \cdots \sigma_1(\mathbf{W}_1 \mathbf{x}) \cdots)), \tag{2.2}$$

where $\sigma_l$ is an activation function and $\boldsymbol{\theta}^* \in \mathbb{R}^d$ is the weight of the output layer. To simplify the presentation, we will assume $\sigma_1 = \sigma_2 = \ldots = \sigma_L = \sigma$ is the ReLU activation function, i.e., $\sigma(x) = \max\{0, x\}$ for $x \in \mathbb{R}$. We denote $\mathbf{w} = (\mathrm{vec}(\mathbf{W}_1)^\top, \ldots, \mathrm{vec}(\mathbf{W}_L)^\top)^\top$, which is the concatenation of the vectorized weight parameters of all hidden layers of the neural network. We also write $f(\mathbf{x}; \boldsymbol{\theta}^*, \mathbf{w}) = f(\mathbf{x})$ in order to explicitly specify the weight parameters of neural network $f$. It

is easy to show that the dimension $p$ of vector $\mathbf{w}$ satisfies $p = (L-2)m^2 + 2md$. To simplify the notation, we define $\phi(\mathbf{x}; \mathbf{w})$ as the output of the $L$-th hidden layer of neural network $f$.

$$\phi(\mathbf{x}; \mathbf{w}) = \sqrt{m}\sigma(\mathbf{W}_L\sigma(\mathbf{W}_{L-1}\cdots\sigma(\mathbf{W}_1\mathbf{x})\cdots)). \tag{2.3}$$

Note that $\phi(\mathbf{x}; \mathbf{w})$ itself can also be viewed as a neural network with vector-valued outputs.

## 3  Deep Representation and Shallow Exploration

The linear parametric form in linear contextual bandits might produce biased estimates of the reward due to the lack of representation power [42, 38]. In contrast, it is well known that deep neural networks are powerful enough to approximate an arbitrary function [18]. Therefore, a natural extension of linear contextual bandits is to use a deep neural network to approximate the reward generating function $r(\cdot)$. Nonetheless, DNNs usually have a prohibitively large dimension for weight parameters, which makes the exploration in neural networks based UCB algorithm inefficient [28, 52].

In this work, we study a neural contextual bandit algorithm, where the hidden layers of a deep neural network are used to represent the features and the exploration is only performed in the last layer of the neural network. In particular, we assume that the reward generating function $r(\cdot)$ can be expressed as the inner product between a deep represented feature vector and an exploration weight parameter, namely, $r(\cdot) = \langle \boldsymbol{\theta}^*, \boldsymbol{\psi}(\cdot)\rangle$, where $\boldsymbol{\theta}^* \in \mathbb{R}^d$ is some weight parameter and $\boldsymbol{\psi}(\cdot)$ is an unknown feature mapping. This decoupling of the representation and the exploration will achieve the best of both worlds: efficient exploration in shallow (linear) models and high expressive power of deep models. To learn the unknown feature mapping, we propose to use a neural network to approximate it. In what follows, we will describe a neural contextual bandit algorithm that uses the output of the last hidden layer of a neural network to transform the raw feature vectors (*deep representation*) and performs UCB-type exploration in the last layer of the neural network (*shallow exploration*). Since the exploration is performed only in the last linear layer, we call this procedure Neural-LinUCB, which is displayed in Algorithm 1.

Specifically, in round $t$, the agent receives an action set with raw features $\mathcal{X}_t = \{\mathbf{x}_{t,1}, \ldots, \mathbf{x}_{t,K}\}$. Then the agent chooses an arm $a_t$ that maximizes the following upper confidence bound:

$$a_t = \underset{k \in [K]}{\operatorname{argmax}}\left\{\langle \phi(\mathbf{x}_{t,k}; \mathbf{w}_{t-1}), \boldsymbol{\theta}_{t-1}\rangle + \alpha_t \|\phi(\mathbf{x}_{t,k}; \mathbf{w}_{t-1})\|_{\mathbf{A}_{t-1}^{-1}}\right\}, \tag{3.1}$$

where $\boldsymbol{\theta}_{t-1}$ is a point estimate of the unknown weight in the last layer, $\phi(\mathbf{x}; \mathbf{w})$ is defined as in (2.3), $\mathbf{w}_{t-1}$ is an estimate of all the weight parameters in the hidden layers of the neural network, $\alpha_t > 0$ is the algorithmic parameter controlling the exploration, and $\mathbf{A}_t$ is a matrix defined based on historical transformed features:

$$\mathbf{A}_t = \lambda\mathbf{I} + \sum_{i=1}^{t}\phi(\mathbf{x}_{i,a_i}; \mathbf{w}_{i-1})\phi(\mathbf{x}_{i,a_i}; \mathbf{w}_{i-1})^\top, \tag{3.2}$$

and $\lambda > 0$. After pulling arm $a_t$, the agent will observe a noisy reward $\widehat{r}_t := \widehat{r}(\mathbf{x}_{t,a_t})$ defined as

$$\widehat{r}(\mathbf{x}_{t,k}) = r(\mathbf{x}_{t,k}) + \xi_t, \tag{3.3}$$

where $\xi_t$ is an independent $\nu$-subGaussian random noise for some $\nu > 0$ and $r(\cdot)$ is an unknown reward function. In this paper, we will interchangeably use notation $\widehat{r}_t$ to denote the reward received at the $t$-th step and an equivalent notation $\widehat{r}(\mathbf{x})$ to express its dependence on the feature vector $\mathbf{x}$.

Upon receiving the reward $\widehat{r}_t$, the agent updates its estimate $\boldsymbol{\theta}_t$ of the output layer weight by using the same $\ell^2$-regularized least-squares estimate in linear contextual bandits [1]. In particular, we have

$$\boldsymbol{\theta}_t = \mathbf{A}_t^{-1}\mathbf{b}_t, \tag{3.4}$$

where $\mathbf{b}_t = \sum_{i=1}^{t}\widehat{r}_i\phi(\mathbf{x}_{i,a_i}; \mathbf{w}_{i-1})$.

To save the computation, the neural network $\phi(\cdot; \mathbf{w}_t)$ will be updated once every $H$ steps. Therefore, we have $\mathbf{w}_{(q-1)H+1} = \ldots = \mathbf{w}_{qH}$ for $q = 1, 2, \ldots$. We call the time steps $\{(q-1)H+1, \ldots, qH\}$ an epoch with length $H$. At time step $t = Hq$, for any $q = 1, 2, \ldots$, Algorithm 1 will retrain the

neural network based on all the historical data. In Algorithm 2, our goal is to minimize the following empirical loss function:

$$\mathcal{L}_q(\mathbf{w}) = \sum_{i=1}^{qH} \left( \boldsymbol{\theta}_i^\top \phi(\mathbf{x}_{i,a_i}; \mathbf{w}) - \widehat{r}_i \right)^2. \tag{3.5}$$

In practice, one can further save computational cost by only feeding data $\{\mathbf{x}_{i,a_i}, \widehat{r}_i, \boldsymbol{\theta}_i\}_{i=(q-1)H+1}^{qH}$ from the $q$-th epoch into Algorithm 2 to update the parameter $\mathbf{w}_t$, which does not hurt the performance since the historical information has been encoded into the estimate of $\boldsymbol{\theta}_i$. In this paper, we will perform the following gradient descent step

$$\mathbf{w}_q^{(s)} = \mathbf{w}_q^{(s-1)} - \eta_q \nabla_{\mathbf{w}} \mathcal{L}_q(\mathbf{w}^{(s-1)}).$$

for $s = 1, \ldots, n$, where $\mathbf{w}_q^{(0)} = \mathbf{w}^{(0)}$ is chosen as the same random initialization point. We will discuss more about the initial point $\mathbf{w}^{(0)}$ in the next paragraph. Then Algorithm 2 outputs $\mathbf{w}_q^{(n)}$ and we set it as the updated weight parameter $\mathbf{w}_{Hq+1}$ in Algorithm 1. In the next round, the agent will receive another action set $\mathcal{X}_{t+1}$ with raw feature vectors and repeat the above steps to choose the sub-optimal arm and update estimation for contextual parameters.

**Initialization:** Recall that $\mathbf{w}$ is the collection of all hidden layer weight parameters of the neural network. We will follow the same initialization scheme as used in Zhou et al. [52], where each entry of the weight matrices follows some Gaussian distribution. Specifically, for any $l \in \{1, \ldots, L-1\}$, we set $\mathbf{W}_l = \begin{bmatrix} \mathbf{W} & \mathbf{0} \\ \mathbf{0} & \mathbf{W} \end{bmatrix}$, where each entry of $\mathbf{W}$ follows distribution $N(0, 4/m)$ independently; for $\mathbf{W}_L$, we set it as $[\mathbf{V} \quad -\mathbf{V}]$, where each entry of $\mathbf{V}$ follows distribution $N(0, 2/m)$ independently.

**Comparison with LinUCB and NeuralUCB:** Compared with linear contextual bandits in Section 2.1, Algorithm 1 has a distinct feature that it learns a deep neural network to obtain a deep representation of the raw data vectors and then performs UCB exploration. This deep representation allows our algorithm to characterize more intrinsic and latent information about the raw data $\{\mathbf{x}_{t,k}\}_{t \in [T], k \in [K]} \subset \mathbb{R}^d$. However, the increased complexity of the feature mapping $\phi(\cdot; \mathbf{w})$ also introduces great hardness in training. For instance, a recent work by Zhou et al. [52] also studied the neural contextual bandit problem, but different from (3.1), their algorithm (NeuralUCB) performs the UCB exploration on the entire network parameter space, which is $\mathbb{R}^{\widetilde{p}+d}$, where $\widetilde{p} = m + md + (L-1)m^2$. Note that in Zhou et al. [52], they need to compute the inverse of a matrix $\mathbf{Z}_t \in \mathbb{R}^{(\widetilde{p}+d) \times (\widetilde{p}+d)}$, which is defined in a similar way to the matrix $\mathbf{A}_t$ in our paper except that $\mathbf{Z}_t$ is defined based on the gradient of the network instead of the output of the last hidden layer as in (3.2). In sharp contrast, $\mathbf{A}_t$ in our paper is only of size $d \times d$ and thus is much more efficient and practical in implementation, which will be seen from our experiments in later sections.

We note that there is also a similar algorithm to our Neural-LinUCB presented in Deshmukh et al. [20], where they studied the self-supervised learning loss in contextual bandits with neural network representation for computer vision problems. However, no regret analysis has been provided. When the feature mapping $\phi(\cdot; \mathbf{w})$ is an identity function, the problem reduces to linear contextual bandits where we directly use $\mathbf{x}_t$ as the feature vector. In this case, it is easy to see that Algorithm 1 reduces to LinUCB [16] since we do not need to learn the representation parameter $\mathbf{w}$ anymore.

**Comparison with Neural-Linear:** The high-level idea of decoupling the representation and exploration in our algorithm is also similar to that of the Neural-Linear algorithm [38, 49], which trains a deep neural network to learn a representation of the raw feature vectors, and then uses a Bayesian linear regression to estimate the uncertainty in the bandit problem. However, these two algorithms are significantly different since Neural-Linear [38] is a Thompson sampling based algorithm that uses posterior sampling to estimate the weight parameter $\boldsymbol{\theta}^*$ via Bayesian linear regression, whereas Neural-LinUCB adopts upper confidence bound based techniques to estimate the weight $\boldsymbol{\theta}^*$. Nevertheless, both algorithms share the same idea of deep representation and shallow exploration, and we view our Neural-LinUCB algorithm as one instantiation of the Neural-Linear scheme.

## 4 Main Results

To analyze the regret bound of Algorithm 1, we first lay down some important assumptions on the neural contextual bandit model.

---
**Algorithm 1** Deep Representation and Shallow Exploration (Neural-LinUCB)

---
1: **Input**: regularization parameter $\lambda > 0$, number of total steps $T$, episode length $H$, exploration parameters $\{\alpha_t > 0\}_{t \in [T]}$
2: **Initialization:** $\mathbf{A}_0 = \lambda \mathbf{I}$, $\mathbf{b}_0 = \mathbf{0}$; entries of $\boldsymbol{\theta}_0$ follow $N(0, 1/d)$, and $\mathbf{w}^{(0)}$ is initialized as described in Section 3; $q = 1$; $\mathbf{w}_0 = \mathbf{w}^{(0)}$
3: **for** $t = 1, \ldots, T$ **do**
4:     receive feature vectors $\{\mathbf{x}_{t,1}, \ldots, \mathbf{x}_{t,K}\}$
5:     choose arm $a_t = \mathrm{argmax}_{k \in [K]} \, \boldsymbol{\theta}_{t-1}^{\top} \boldsymbol{\phi}(\mathbf{x}_{t,k}; \mathbf{w}_{t-1}) + \alpha_t \|\boldsymbol{\phi}(\mathbf{x}_{t,k}; \mathbf{w}_{t-1})\|_{\mathbf{A}_{t-1}^{-1}}$, and obtain reward $\widehat{r}_t$
6:     update $\mathbf{A}_t$ and $\mathbf{b}_t$ as follows:
      $\mathbf{A}_t = \mathbf{A}_{t-1} + \boldsymbol{\phi}(\mathbf{x}_{t,a_t}; \mathbf{w}_{t-1}) \boldsymbol{\phi}(\mathbf{x}_{t,a_t}; \mathbf{w}_{t-1})^{\top}$,
      $\mathbf{b}_t = \mathbf{b}_{t-1} + \widehat{r}_t \boldsymbol{\phi}(\mathbf{x}_{t,a_t}; \mathbf{w}_{t-1})$,
7:     update $\boldsymbol{\theta}_t = \mathbf{A}_t^{-1} \mathbf{b}_t$
8:     **if** $\mathrm{mod}(t, H) = 0$ **then**
9:       $\mathbf{w}_t \leftarrow$ output of Algorithm 2
10:       $q = q + 1$
11:     **else**
12:       $\mathbf{w}_t = \mathbf{w}_{t-1}$
13:     **end if**
14: **end for**
15: **Output** $\mathbf{w}_T$

---

---
**Algorithm 2** Update Weight Parameters with Gradient Descent

---
1: **Input:** initial point $\mathbf{w}_q^{(0)} = \mathbf{w}^{(0)}$, maximum iteration number $n$, step size $\eta_q$, and loss function defined in (3.5).
2: **for** $s = 1, \ldots, n$ **do**
3:     $\mathbf{w}_q^{(s)} = \mathbf{w}_q^{(s-1)} - \eta_q \nabla_{\mathbf{w}} \mathcal{L}_q(\mathbf{w}_q^{(s-1)})$.
4: **end for**
5: **Output** $\mathbf{w}_q^{(n)}$

---

**Assumption 4.1.** For all $i \geq 1$ and $k \in [K]$, we assume that $\|\mathbf{x}_{i,k}\|_2 = 1$ and its entries satisfy $[\mathbf{x}_{i,k}]_j = [\mathbf{x}_{j,k}]_{j+d/2}$.

The assumption that $\|\mathbf{x}_{i,k}\|_2 = 1$ is not essential and is only imposed for simplicity, which is also used in Zou and Gu [53], Zhou et al. [52]. Finally, the condition on the entries of $\mathbf{x}_{i,k}$ is also mild since otherwise we could always construct $\mathbf{x}'_{i,k} = [\mathbf{x}_{i,k}^{\top}, \mathbf{x}_{i,k}^{\top}]^{\top} / \sqrt{2}$ to replace it. An implication of Assumption 4.1 is that the initialization scheme in Algorithm 1 results in $\boldsymbol{\phi}(\mathbf{x}_{i,k}; \mathbf{w}^{(0)}) = \mathbf{0}$ for all $i \in [T]$ and $k \in [K]$.

We assume the following stability condition on the spectral norm of the neural network gradient:

**Assumption 4.2.** There is a constant $\ell_{\mathrm{Lip}} > 0$ such that it holds

$$\left\| \frac{\partial \boldsymbol{\phi}}{\partial \mathbf{w}}(\mathbf{x}; \mathbf{w}_0) - \frac{\partial \boldsymbol{\phi}}{\partial \mathbf{w}}(\mathbf{x}'; \mathbf{w}_0) \right\|_2 \leq \ell_{\mathrm{Lip}} \|\mathbf{x} - \mathbf{x}'\|_2,$$

for all $\mathbf{x}, \mathbf{x}' \in \{\mathbf{x}_{i,k}\}_{i \in [T], k \in [K]}$.

The inequality in Assumption 4.2 resembles the Lipschitz condition on the gradient of the neural network. However, it is essentially different from the smoothness condition since here the gradient is taken with respect to the neural network weights while the Lipschitz condition is imposed on the feature parameter $\mathbf{x}$. Similar conditions are widely made in nonconvex optimization [46, 10, 48], in the name of first-order stability, which is essential to derive the convergence of alternating optimization algorithms. Furthermore, Assumption 4.2 is only required on the $TK$ training data points and a specific weight parameter $\mathbf{w}_0$. Therefore, the condition will hold if the raw feature data lie in a certain subspace of $\mathbb{R}^d$. We provided some further discussions in the supplementary material about this assumption for interested readers.

In order to analyze the regret bound of Algorithm 1, we need to characterize the properties of the deep neural network in (2.2) that is used to represent the feature vectors. Following a recent line of research [27, 12, 7, 52], we define the covariance between two data point $\mathbf{x}, \mathbf{y} \in \mathbb{R}^d$ as follows.

$$\widetilde{\boldsymbol{\Sigma}}^{(0)}(\mathbf{x}, \mathbf{y}) = \boldsymbol{\Sigma}^{(0)}(\mathbf{x}, \mathbf{y}) = \mathbf{x}^\top \mathbf{y},$$

$$\boldsymbol{\Lambda}^{(l)}(\mathbf{x}, \mathbf{y}) = \begin{bmatrix} \boldsymbol{\Sigma}^{l-1}(\mathbf{x}, \mathbf{x}) & \boldsymbol{\Sigma}^{l-1}(\mathbf{x}, \mathbf{y}) \\ \boldsymbol{\Sigma}^{l-1}(\mathbf{y}, \mathbf{x}) & \boldsymbol{\Sigma}^{l-1}(\mathbf{y}, \mathbf{y}) \end{bmatrix},$$

$$\boldsymbol{\Sigma}^{(l)}(\mathbf{x}, \mathbf{y}) = 2\mathbb{E}_{(u,v) \sim N(\mathbf{0}, \boldsymbol{\Lambda}^{(l-1)}(\mathbf{x}, \mathbf{y}))}[\sigma(u)\sigma(v)],$$

$$\widetilde{\boldsymbol{\Sigma}}^{(l)}(\mathbf{x}, \mathbf{y}) = 2\widetilde{\boldsymbol{\Sigma}}^{(l-1)}(\mathbf{x}, \mathbf{y})\mathbb{E}_{u,v}[\dot{\sigma}(u)\dot{\sigma}(v)] + \boldsymbol{\Sigma}^{(l)}(\mathbf{x}, \mathbf{y}), \tag{4.1}$$

where $(u, v) \sim N(\mathbf{0}, \boldsymbol{\Lambda}^{(l-1)}(\mathbf{x}, \mathbf{y}))$, and $\dot{\sigma}(\cdot)$ is the derivative of activation function $\sigma(\cdot)$. We denote the neural tangent kernel (NTK) matrix $\mathbf{H} \in \mathbb{R}^{TK \times TK}$ based on all feature vectors $\{\mathbf{x}_{t,k}\}_{t \in [T], k \in [K]}$. Renumbering $\{\mathbf{x}_{t,k}\}_{t \in [T], k \in [K]}$ as $\{\mathbf{x}_i\}_{i=1,\ldots,TK}$, then each entry $\mathbf{H}_{ij}$ is defined as

$$\mathbf{H}_{ij} = \frac{1}{2}\big(\widetilde{\boldsymbol{\Sigma}}^{(L)}(\mathbf{x}_i, \mathbf{x}_j) + \boldsymbol{\Sigma}^{(L)}(\mathbf{x}_i, \mathbf{x}_j)\big), \tag{4.2}$$

for all $i, j \in [TK]$. Based on the above definition, we impose the following assumption on $\mathbf{H}$.

**Assumption 4.3.** The neural tangent kernel defined in (4.2) is positive definite, i.e., $\lambda_{\min}(\mathbf{H}) \geq \lambda_0$ for some constant $\lambda_0 > 0$.

Assumption 4.3 essentially requires the neural tangent kernel matrix $\mathbf{H}$ to be non-singular, which is a mild condition and also imposed in other related work [21, 7, 12, 52]. Moreover, it is shown that Assumption 4.3 can be easily derived from Assumption 4.1 for two-layer ReLU networks [37, 53]. Therefore, Assumption 4.3 is mild or even negligible given the non-degeneration assumption on the feature vectors. Also note that matrix $\mathbf{H}$ is only defined based on layers $l = 1, \ldots, L$ of the neural network, and does not depend on the output layer $\boldsymbol{\theta}$. It is easy to extend the definition of $\mathbf{H}$ to the NTK matrix defined on all layers including the output layer $\boldsymbol{\theta}$, which would also be positive definite by Assumption 4.3 and the recursion in (4.2).

Before we present the regret analysis of the neural contextual bandit, we need to modify the regret defined in (2.1) to account for the randomness of the neural network initialization. For a fixed time horizon $T$, we define the regret of Algorithm 1 as follows.

$$R_T = \mathbb{E}\left[\sum_{t=1}^{T} \big(\widehat{r}(\mathbf{x}_{t,a_t^*}) - \widehat{r}(\mathbf{x}_{t,a_t})\big)\big|\mathbf{w}^{(0)}\right], \tag{4.3}$$

where the expectation is taken over the randomness of the reward noise. Note that $R_T$ defined in (4.3) is still a random variable since the initialization of Algorithm 2 is randomly generated.

Now we are going to present the regret bound of the proposed algorithm.

**Theorem 4.4.** Suppose Assumptions 4.1, 4.2 and 4.3 hold. Assume that $\|\boldsymbol{\theta}^*\|_2 \leq M$ for some positive constant $M > 0$. For any $\delta \in (0, 1)$, let us choose $\alpha_t$ in Neural-LinUCB as

$$\alpha_t = \nu\sqrt{2\big(d\log(1 + t\log(HK)/\lambda) + \log(1/\delta)\big)} + \lambda^{1/2}M.$$

We choose the step size $\eta_q$ of Algorithm 2 as

$$\eta_q \leq C_0\big(d^2mnT^{5.5}L^6\log(TK/\delta)\big)^{-1},$$

and the width of the neural network satisfies $m = \text{poly}(L, d, 1/\delta, H, \log(TK/\delta))$. With probability at least $1 - \delta$ over the randomness of the initialization of the neural network, it holds that

$$R_T \leq C_1\alpha_T\sqrt{Td\log\left(1 + \frac{TG^2}{\lambda d}\right)} + \frac{C_2\ell_{\text{Lip}}L^3d^{5/2}T\sqrt{\log m \log(\frac{1}{\delta})\log(\frac{TK}{\delta})}\|\mathbf{r} - \widetilde{\mathbf{r}}\|_{\mathbf{H}^{-1}}}{m^{1/6}},$$

where $\{C_i\}_{i=0,1,2}$ are absolute constants independent of the problem parameters, $\mathbf{r} = (r(\mathbf{x}_1), r(\mathbf{x}_2), \ldots, r(\mathbf{x}_{TK}))^\top \in \mathbb{R}^{TK}$ and $\widetilde{\mathbf{r}} = (f(\mathbf{x}_1; \boldsymbol{\theta}_0, \mathbf{w}_0), \ldots, f(\mathbf{x}_{TK}; \boldsymbol{\theta}_{T-1}, \mathbf{w}_{T-1}))^\top \in \mathbb{R}^{TK}$, and $\|\mathbf{r}\|_{\mathbf{A}} = \sqrt{\mathbf{r}^\top \mathbf{A}\mathbf{r}}$.

**Remark 4.5.** Theorem 4.4 shows that the regret of Algorithm 1 can be bounded by two parts: the first part is of order $\widetilde{O}(\sqrt{T})$, which resembles the regret bound of linear contextual bandits [1]; the second part is of order $\widetilde{O}(m^{-1/6}T\sqrt{(\mathbf{r}-\widetilde{\mathbf{r}})^\top\mathbf{H}^{-1}(\mathbf{r}-\widetilde{\mathbf{r}})})$, which depends on the estimation error of the neural network $f$ for the reward generating function $r$ and the neural tangent kernel $\mathbf{H}$.

It is worth noting that our theoretical analysis depends on the reward structure assumption that $r(\cdot) = \langle\boldsymbol{\theta}_*, \boldsymbol{\psi}(\cdot)\rangle$. However, the linear structure between $\boldsymbol{\theta}_*$ and $\boldsymbol{\psi}(\cdot)$ is not essential. As long as the deep representation of the feature vector and the uncertainty weight parameter can be decoupled, Algorithm 1 can be easily extended to settings with milder assumptions on the reward structure such as generalized linear models [41, 24, 35, 28]. For more general bandit models where no assumption is imposed to the reward generating function, it is still unclear whether the decoupled deep representation and shallow exploration would work especially in cases a thorough exploration may be needed.

Based on the result in Theorem 4.4, we can easily verify the following conclusion:

**Corollary 4.6.** Under the same conditions of Theorem 4.4, if we choose a sufficiently overparameterized neural network mapping $\phi(\cdot)$ such that $m \geq T^3$, then the regret of Algorithm 1 is $R_T = \widetilde{O}(\sqrt{T}\sqrt{(\mathbf{r}-\widetilde{\mathbf{r}})^\top\mathbf{H}^{-1}(\mathbf{r}-\widetilde{\mathbf{r}})})$.

**Remark 4.7.** For the ease of presentation, let us denote $\mathcal{E} := \|\mathbf{r}-\widetilde{\mathbf{r}}\|_{\mathbf{H}^{-1}}$. If we have $\mathcal{E} = O(1)$, the total regret in Theorem 4.4 becomes $\widetilde{O}(\sqrt{T})$ which matches the regret of linear contextual bandits [1]. We remark that there is a similar assumption in [52] where they assume that $\mathbf{r}^\top\mathbf{H}^{-1}\mathbf{r}$ can be upper bounded by a constant. They show that this term can be bounded by the RKHS norm of $\mathbf{r}$ if it belongs to the RKHS induced by the neural tangent kernel [6, 7, 33]. In addition, $\mathcal{E}$ here is the difference between the true reward function and the neural network function, which can also be small if the deep neural network function well approximates the reward generating function $r(\cdot)$.

## 5  Experiments

In this section, we provide empirical evaluations of Neural-LinUCB on real-world datasets. As we have discussed in Section 3, Neural-LinUCB could be viewed as an instantiation of the Neural-Linear scheme studied in Riquelme et al. [38] except that we use the UCB exploration instead of the posterior sampling exploration therein. Note that there has been an extensive comparison [38] of the Neural-Linear methods with many other baselines such as greedy algorithms, Variational Inference, Expectation-Propagation, Bayesian Non-parametrics and so on. Therefore, we do not seek a thorough empirical comparison of Neural-LinUCB with all existing bandits algorithms. We refer readers who are interested in the performance of Neural-Linear methods with deep representation and shallow exploration compared with a vast of baselines in the literature to the benchmark study by Riquelme et al. [38]. In this experiment, we only aim to show the advantages of our algorithm over the following baselines: (1) Neural-Linear [38]; (2) LinUCB [16], which does not have a *deep representation* of the feature vectors; and (3) NeuralUCB [52], which performs UCB exploration on all the parameters of the neural network instead of the *shallow exploration* used in our paper. All numerical experiments were run on a workstation with Intel(R) Xeon(R) CPU E5-2637 v4 @ 3.50GHz.

**Datasets:** we evaluate the performances of all algorithms on bandit problems created from real-world data. Specifically, following the experimental setting in Zhou et al. [52],we use datasets *(Shuttle) Statlog*, *Magic* and *Covertype* from UCI machine learning repository [23], and the *MINST* dataset from LeCun et al. [31]. The details of these datasets are presented in Table 1. In Table 1, each instance represents a feature vector $\mathbf{x} \in \mathbb{R}^d$ that is associated with one of the $K$ arms, and dimension $d$ is the number of attributes in each instance.

Table 1: Specifications of datasets from the UCI machine learning repository used in this paper.

|                      | *Statlog* | *Magic* | *Covertype* | *MNIST* |
| -------------------- | --------- | ------- | ----------- | ------- |
| Number of attributes | 9         | 11      | 54          | 784     |
| Number of arms       | 7         | 2       | 7           | 10      |
| Number of instances  | 58,000    | 19,020  | 581,012     | 60,000  |

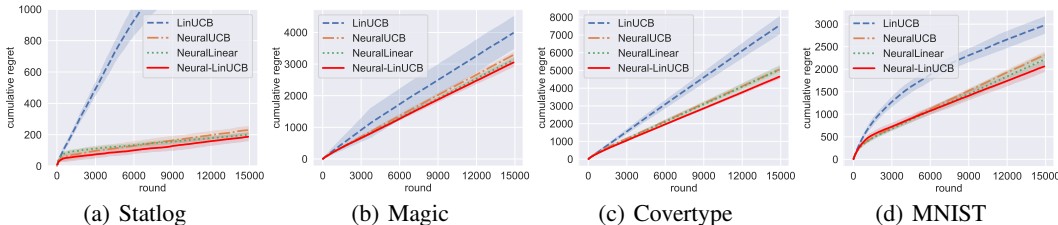

| (a) Statlog | (b) Magic | (c) Covertype | (d) MNIST |

Figure 1: The cumulative regrets of LinUCB, NeuralUCB, Neural-Linear and Neural-LinUCB over 15, 000 rounds. Experiments are averaged over 10 repetitions.

**Implementations:** for LinUCB, we follow the setting in Li et al. [34] to use disjoint models for different arms. For neural network based algorithms such as NeuralUCB, Neural-Linear and Neural-LinUCB, we use a ReLU neural network defined as in (2.2) with $L = 2$ and 2000 for the UCI datasets (*Statlog*, *Magic*, *Covertype*). Thus the neural network weights are $\mathbf{W}_1 \in \mathbb{R}^{m \times d}$, $\mathbf{W}_2 \in \mathbb{R}^{k \times m}$, and $\boldsymbol{\theta} \in \mathbb{R}^k$ respectively, where $k = 100$, $m = 2000$, and $d$ is the dimension of features in the corresponding task. Since the problem size of the MNIST dataset is larger, inspired by Hinton and Salakhutdinov [26], we use a deeper NN and set $L = 3$, $k = 100$ and $m = 100$, with weights $\mathbf{W}_1 \in \mathbb{R}^{m \times d}$, $\mathbf{W}_2 \in \mathbb{R}^{m \times m}$, $\mathbf{W}_3 \in \mathbb{R}^{k \times m}$, and $\boldsymbol{\theta} \in \mathbb{R}^k$. We set the time horizon $T = 15,000$, which is the total number of rounds for each algorithm on each dataset. We use gradient decent to optimize the network weights, with a step size $\eta_q =$1e-5 and maximum iteration number $n = 1,000$. To speed up the training process, the network parameter $\mathbf{w}$ is updated every $H = 100$ rounds starting from round 2000. We also apply early stopping when the loss difference of two consecutive iterations is smaller than a threshold of 1e-6. We set $\lambda = 1$ and $\alpha_t = 0.02$ for all algorithms, $t \in [T]$. Following the setting in Riquelme et al. [38], we use round-robin to independently select each arm for 3 times at the beginning of each algorithm. For NeuralUCB, since it is computationally unaffordable to perform the original UCB exploration as displayed in Zhou et al. [52], we follow their experimental setting to replace the matrix $\mathbf{Z}_t \in \mathbb{R}^{(d+\widetilde{p}) \times (d+\widetilde{p})}$ in Zhou et al. [52] with its diagonal matrix.

**Results:** we plot the cumulative regret of all algorithms versus round in Figures 1(a), 1(b) and 1(c) for UCI datasets and in Figure 1(d) for MNIST. The results are reported based on the average of 10 repetitions over different random shuffles of the datasets. It can be seen that algorithms based on neural network representations (NeuralUCB, Neural-Linear and Neural-LinUCB) consistently outperform the linear contextual bandit method LinUCB, which shows that linear models may lack representation power and find biased estimates for the underlying reward generating function. Furthermore, our proposed Neural-LinUCB achieves a comparable regret with NeuralUCB in all experiments despite the fact that our algorithm only explores in the output layer of the neural network, which is more computationally efficient as we will show in the sequel.The results in our experiment are well aligned with our theory that deep representation and shallow exploration are sufficient to guarantee a good performance of neural contextual bandit algorithms, which is also consistent with the findings in existing literature [38] that decoupling the representation learning and uncertainty estimation improves the performance.

We also conducted experiments to study the effects of different widths of deep neural networks on the regret performance and to show the computational efficiency of Neural-LinUCB compared with existing neural bandit algorithms. Due to the space limit, we defer the results to Appendix A.

## 6 Conclusions

In this paper, we propose a new neural contextual bandit algorithm called Neural-LinUCB, which uses the hidden layers of a ReLU neural network as a deep representation of the raw feature vectors and performs UCB type exploration on the last layer of the neural network. By incorporating techniques in liner contextual bandits and neural tangent kernels, we prove that the proposed algorithm achieves a sublinear regret when the width of the network is sufficiently large. This is the first regret analysis of neural contextual bandit algorithms with deep representation and shallow exploration, which have been observed in practice to work well on many benchmark bandit problems [38]. We also conducted experiments on real-world datasets to demonstrate the advantage of the proposed algorithm over LinUCB and existing neural contextual bandit algorithms.

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
