# OpenReview forum: "Neural Contextual Bandits with Deep Representation and Shallow Exploration"
_NeurIPS.cc/2021/Conference — NeurIPS 2021 Submitted_

### Official Review · Reviewer_1Vq8 · 2021-07-13

**Rating:** 7
**Confidence:** 3

**Summary:**

This paper proposes a novel contextual bandit algorithm, Neural-LinUCB, that transforms the raw feature vector using the last hidden layer of a deep ReLU neural network (deep representation learning), and uses an upper confidence bound (UCB) approach to explore in the last linear layer (shallow exploration). The algorithm is shown to enjoy the best of two worlds: strong expressiveness due to the deep representation and computational efficiency due to the shallow exploration. In theory, a $\sqrt{T}$ regret bound is established. Numerical experiments further demonstrated the importance of decoupling representation learning and uncertainty estimation.

**Main Review:**

Overall, this paper is very well written and is easy to follow. The contribution over NeuralUCB (Zhou et al. 2020) is clear and convincing. Below are some questions for further clarification.

1. The algorithm relies on the exploration parameter $\alpha_t$, which is set to be an increasing function of $t$ and an unknown constant $M$ in Theorem 4.4. However, in the experiment, $\alpha_t = 0.02$ for all $t$. Can authors provide any justification or explanations on this inconsistency.

2. As shown in Corollary 4.6, the final rate is $\sqrt{T}$ only when $\sqrt{(r-\tilde{r})^{\top} H^{-1} (r-\tilde{r})}$ is bounded. Note that $r \in R^{TK}$ and $H\in R^{TK \times TK}$. The authors justified this by claiming that Zhou et al. (2020) assumed that this can be upper bounded by a constant. Personally, I would like to see more justifications on this statement as the $\sqrt{T}$ regret bound was claimed in the abstract and introduction.

3. Suggestion: one advantage of the proposed Neural-LinUCB over NeuralUCB (Zhou et al. 2020) is the computational efficiency. I would suggest the authors to move some runtime comparisons in Section A.1 of the supplement to the main paper.

4. Typo: line 313, "$L=2$ and $2000$" should be "$L=2$ and $m=2000$".


**Time Spent Reviewing:**

5

---

> ### Author Response · Authors · 2021-08-10
> **Response to Reviewer 1Vq8**
>
> Thank you for your supportive comments. We will fix the typo in the camera ready. We address your comments point by point as follows.
>
> ---
> **Q**: “In Theorem 4.4, the exploration parameter is an increasing function of  t and an unknown constant M. However, in the experiment, $\alpha=0.02$ for all t.”
> **A**: Theoretically, the exploration parameter $\alpha_t$ does not need to be an increasing function. We can simply replace $\alpha_t$ with its upper bound $\alpha_T$ for all $t$ and the proofs do not need much modification. In practice, the exploration parameter is commonly chosen as a constant or treated as a tuning parameter in experiments in the literature.
>
> ---
> **Q**: “More justifications on the statement that $\sqrt{(r-\tilde{r})^{\top} H^{-1} (r-\tilde{r})}$ can be assumed to be bounded”
> **A**: We would like to point out that Zhou et al. (2020) proved that $\sqrt{r^{\top} H^{-1} r}$ is upper bounded when the reward generating function $r$ lies in the RKHS induced by the neural tangent kernel. Note that the term in Corollary 4.6 of our paper depends on the approximation error (or the residual) $\sqrt{(r-\tilde{r})^{\top} H^{-1} (r-\tilde{r})}$, which is much smaller than $\sqrt{r^{\top} H^{-1} r}$ if the neural network well approximates the reward function. Therefore, it is reasonable to assume the residual to be bounded by a constant as well.
>
> ---
> **Q**: “Move some runtime comparisons in Section A.1 of the supplement to the main paper”
> **A**: Thank you for the great suggestion. We will move it to the main paper in the camera ready.

---

### Official Review · Reviewer_LYF3 · 2021-07-15

**Rating:** 6
**Confidence:** 4

**Summary:**

The paper studies the contextual bandit without knowing the specific reward generating function. The authors propose to use a deep neural network to model the reward function where the hidden layers of the deep neural network are used to represent the features and the exploration is only performed in the last layer of the
neural network. In this way, the reward generating function can be expressed as the inner product between a deeply represented feature vector and an exploration weight parameter. As the author explained, the main advantage of this reward model is to efficiently reduce computation because we only need to explore the last layer of the deep neural network instead of exploring the whole network like existing neural bandit algorithms.
The authors show that the proposed algorithm for this problem obtains a near-optimal regret which matches the regret bound of linear contextual bandits. They also provided experiments based on real-world datasets to show the performance and computational efficiency.


**Limitations And Societal Impact:**

Yes

**Main Review:**

The idea of using a deep representation and an exploration at the last layer of a neural network has been exploited in the literature. I think that their main contribution consists of a novel algorithm using the UCB approach and their theoretical results. The paper sounds technically correct.
Overall, the paper is very well written and easy to follow. It would be better if the authors can clearly explain the notation of “exploration” used in the last layer. I understand that this notation is different from that of the exploitation-exploration tradeoff in the contextual bandit.
I have several questions as follows:
To guarantee the near-optimal regret, the neural network must be over-parameterized. As Corollary 4.6 showed, m > T^3 is needed. If T =1000, then m > 10^9. Such a network is so large to implement and train. How did you solve this issue in your experiments and what is a potential solution?
In the experimental results in Figure 1, I see that Neural-LinUCB always outperforms NeuralUCB on all the datasets. Why? Is it because of the computational issue of the inverse matrix Z^{-1} of NeuralUCB?
In contextual bandit, Thompson Sampling algorithms often outperform UCB-based algorithms. See paper “Neural Thompson Sampling”. Why does your UCB-based algorithm always outperform Neural-Linear, a Thompson-Sampling-based algorithm?


**Time Spent Reviewing:**

3

---

> ### Author Response · Authors · 2021-08-10
> **Response to Reviewer LYF3**
>
> Thank you for your constructive comments. We address your comments point by point as follows.
>
> ---
> **Q**: “Explain the notation of ‘exploration’ used in the last layer”
> **A**: When we say exploration in the last layer, we mean after training the neural network up to the last hidden layer, we treat the output of the last hidden layer $\phi(\mathbf{x};\mathbf{w})$ as a learned feature vector. We then apply the UCB strategy to this learned feature vector, which is of the same dimension as the output of the last hidden layer, instead of an extremely high dimensional vector $\text{vec}(\mathbf{w})$ (the whole network weight) as in [51, 52]. This is called shallow exploration in our paper. We will add more explanation of this term in the camera ready version.
>
> ---
> **Q**: “Such a network ($m>T^3$) is so large to implement and train. How did you solve this issue in your experiments and what is a potential solution”
> **A**: We would like to point out that this is just the theoretical value, which is a sufficient condition to prove our results. We believe it is not necessary in practice. In other words, our theory may be conservative since our current understanding of deep learning is still limited. Nevertheless, we believe our work is a good starting point towards understanding the behavior of deep bandits algorithms. Therefore, we only choose moderately wide neural networks (e.g., $m=100$) in our experiments as shown in Section 5.
>
> ---
> **Q**: “Why is Neural-LinUCB better than NeuralUCB? Is it because of the computational issue of the inverse matrix Z^{-1} of NeuralUCB?”
> **A**: Yes, you are right. Due to the computational cost of NeuralUCB (it needs to compute the inverse of a $p\times p$ matrix, where $p=(L-1)m^2+md+m+d$ is extremely large), it only works when a diagonal matrix approximation is used in the experiments, which might lose information about the context vectors.
>
> ---
> **Q**: “Why does your UCB-based algorithm outperform Neural-Linear, a Thompson-Sampling-based algorithm?”
> **A**: First of all, we would like to point out that the slightly worse performance of Neural-Linear than that of Neural-LinUCB is probably not due to Thompson sampling being worse than UCB, but due to the overall algorithm design. The Neural-Linear algorithm provided in Riquelme et al. (2018) is only based on a generic form of Thompson sampling. There are no provable guarantees for Neural-Linear to perform well or even just to achieve a sublinear regret. In contrast, our algorithm Neural-LinUCB is carefully designed such that we for the first time show when and how the high-level idea of decoupling representation and exploration works in both theory and practice. Therefore, we think it is not very surprising that our algorithm is better than Neural-Linear. Second, even when the algorithms are rigorously designed like the case of NeuralUCB and NeuralTS, NeuralTS is not always better than NeuralUCB. See Figure 3(a) and Figure 3(b) in [51] (Neural Thompson Sampling) for details.

---

### Official Review · Reviewer_M9Rm · 2021-07-17

**Rating:** 7
**Confidence:** 2

**Summary:**

Authors provide a new regret bound for a recent approach to solving contextual bandits. Previous approaches have introduced deep neural networks to learn context, leading to a paradigm where one must explore over the entire network parameter space, which is inefficient for learning. The authors address this by taking an existing approach that decouples the deep neural network feature representation learning from most of the exploration of the network parameters by only exploring over the final layer of the network. Although this has been done previously in the context of Thompson sampling, there has not been a regret bound given. The authors analyze a UCB version of this approach, then build from techniques from both deep neural contextual bandits and linear contextual bandits to prove an O(\sqrt(T)) regret bound. Finally, they show experimentally that their algorithm is better than neural-only or linear-only contextual bandits.

**Limitations And Societal Impact:**

yes

**Main Review:**


Strengths:
- Authors combine known techniques to create an interesting algorithm for contextual bandits. Although the combination is not novel, the authors make a theoretical contribution by demonstrating that the combination has O(\sqrt(T)) regret.
- The paper is well written, limitations clearly addressed, and assumptions made clear.
- Authors give results from 3 tabular and 1 image domain, providing a decent test bed for their algorithm against competitors under different neural network depths. Their algorithm always outperforms the competitors.


Weaknesses:
- The way the paper is written somewhat undersells the novelty of the authors' contribution. Right now, their contribution is framed such that their algorithm is a subset of the Neural-linear idea from Riquelme et al., and the purpose of the whole paper is mainly to provide this regret bound for one version of a neural-linear contextual bandit. This left me wanting more -- e.g., are there new conceptual ideas here, were there specific challenges to achieving the proof that could be part of the authors' contribution, does the proof provide some more general implication about the way contextual or other bandits can be learned? Similarly the experimental results showed very little separation between the competitor NeuralUCB and the authors' Neural-LinUCB -- one thing that would have been interesting to see would be an analysis of whether more separation could be achieved between the two, as the size of the networks increase -- that would be an interesting way to experimentally verify the differences between the O(d\sqrt(T)) regret bound of NeuralUCB and the O(\sqrt(T)) regret bound of the authors' method.
---------------------------------------------
Update: I have read the other reviews and the author responses. First, I think the authors adequately addressed my concern (also mentioned by reviewer 7ZGH) that the technical challenges over previous literature needed to reach their results were unclear. Authors should make this clear in their final version. Second, authors make a reasonable argument about why running experiments with larger architectures (where they would see more separation against the competitor) were not carried out. However, some version of this experiment really should be included in the final paper. Authors should at least run a small version of this scaling experiment against the competitor (e.g., network widths of 2000, 3000, 4000...) and include it in the main text (they could be added to Figure 1 relatively easily.) I have increased my score by 1 to reflect the authors addressing my main concerns.

**Time Spent Reviewing:**

2

---

> ### Author Response · Authors · 2021-08-10
> **Response to Reviewer M9Rm**
>
> Thank you for your constructive comments. We address your comments point by point as follows.
>
> ---
> **Q**: “Are there new conceptual ideas, specific challenges in proofs, and general implication about other contextual bandits?”
> **A**: Although the high-level idea of shallow exploration is the same as that in Riquelme et al. (2018), the details of the algorithm design are very different. Apart from the difference between Thompson sampling and UCB based algorithms, their algorithm Neural-Linear is more like a heuristic method without any specific design (such as how to choose and update the posterior distribution) that guarantees the convergence of the algorithm. Our Neural-LinUCB algorithm is carefully designed such that we for the first time show when and how the high-level idea of decoupling representation and exploration works in both theory and practice. Our proof could not be easily adapted from existing analysis of linear contextual bandits or neural bandits since all of them rely on the full exploration of all weight parameters. We believe that our proof techniques can also be used to analyze the provable regret bound of Thompson sampling based algorithms with shallow exploration, and other variants of bandits such as cascading bandits, semi-bandits, etc.
>
> ---
> **Q**: “Could more separation be achieved between NeuralUCB and Neural-LinUCB, as the size of the networks increase?”
> **A**: Thank you for your suggestion. We indeed conducted an experiment in Appendix A to show that the performance of Neural-LinUCB will increase as the network size increases. However, NeuralUCB is hard to scale to large networks even just using the diagonal matrix in the UCB bonus term (the experiments in the original NeuralUCB paper only use two-layer networks with width $m=100$). Nevertheless, given that Neural-LinUCB outperforms NeuralUCB as shown in Figure 1 and that the performance of Neural-LinUCB increases as the network size increases as shown in Figure 2, we believe that the separation between NeuralUCB and Neural-LinUCB will be larger when the network size increase.

---

### Official Review · Reviewer_7ZGH · 2021-07-27

**Rating:** 5
**Confidence:** 5

**Summary:**

The paper studies a contextual bandit problem for which the authors propose a neural network-based policy that takes a raw feature vector as an input without knowledge of the specific reward generating function. The proposed policy transforms the raw feature vector using the last hidden layer of a deep ReLU neural network and uses the upper confidence bound (UCB) approach to explore in the last linear layer which the authors call "shallow exploration" instead of using the entire network to explore. The paper provides regret analysis which shows O(\sqrt{T}) regret. The proposed method enjoys computational advantages over existing neural contextual bandit policies.

**Main Review:**

The authors argue that the proposed decoupling of the representation and the exploration will achieve the best of both worlds: efficient exploration in shallow (linear) models and the high expressive power of deep models. However,  the high-level idea of decoupling the representation and exploration has already been proposed in previous literature [38] although those works focused on a Thompson sampling-based algorithm. Hence, I am not convinced that credit should be given to this paper for the decoupling idea. I believe the paper should be evaluated on theoretical merit and wish that the authors are more explicitly clear about what is technically challenging and what technical innovations they are providing to the community compared to the existing techniques in [51,52].

In Line 262, the authors state that the width of the neural network satisfies m = poly(L, d, 1/δ, H, log(T K/δ)). Is it the same order of m as shown in the previous works [51,52] which is too large (hence very restrictive) for any practical settings? I wish the authors are more clear about this. I feel that what the authors are appealing for is a more practical version of neural contextual bandits with "shallow exploration" But, not stating what is crucial information for their result to hold, which does hinder practicality, is clearly not helping the readers.

While the numerical results show that the performance of the proposed method is slightly better than the existing methods. The plots are not showing any sublinear regret in Figure 1 which is a bit disappointing despite the claim of achieving the best of both worlds.



**Time Spent Reviewing:**

4

---

> ### Author Response · Authors · 2021-08-10
> **Response to Reviewer 7ZGH**
>
> Thank you for your helpful comments. We address your comments point by point as follows.
>
> ---
> **Q**: “The high-level idea of decoupling the representation and exploration has already been proposed in previous literature”
> **A**: First of all, we would like to clarify that one of our contributions is indeed the algorithm design. It is true that the high-level idea of combining deep representation and shallow exploration first appeared in Riquelme et al. (2018). However, their algorithm Neural-Linear is more like a heuristic method without any specific design (such as how to choose and update the posterior distribution) that guarantees the convergence of the algorithm. Our Neural-LinUCB algorithm is carefully designed such that we for the first time show when and how the high-level idea of decoupling representation and exploration works in both theory and practice.
>
> ---
> **Q**: “More explicitly clear about the technical challenge and technical innovations”
> **A**: Thank you for the suggestion. The algorithms in [51, 52] treat the weight of the neural network as a whole, which could be seen as a straightforward generalization of LinUCB and LinTS. In contrast, Neural-LinUCB in our paper decouples the deep representation learning from the UCB exploration. This decoupling imposes more challenges in the analysis since we need to separately show that (1) the network parameter up to the last hidden layer converges; and (2) the last layer parameter is in a confidence ball of the true parameter. No existing theoretical analysis can deal with the same setting in our paper since all of them rely on the full exploration of all weight parameters.
>
> ---
> **Q**: “Is the order of $m$ the same as shown in the previous works [51,52]?”
> **A**: The requirement of the width in our paper ($m=O(T^3)$) is slightly milder than that in [51,52], which is $m=O(T^7)$.
>
> ---
> **Q**: “The width of the neural network is too large; what is crucial for the shallow exploration to work in practice”
> **A**: Recall that the width of the network is $m$. We would like to point out that the requirement on the width ($m=O(T^3)$) in our theorem is only a sufficient condition. In other words, such a requirement may be very conservative and could be further relaxed in practice. In fact, throughout our Neural-LinUCB algorithm, we only need to perform UCB exploration on the output layer, which is a $d$-dimensional vector. Therefore it is practical to calculate the matrix inverse $A_t^{-1}$ in Line 5 and Line 7 of Algorithm 1 as $A_t\in\mathbf{R}^{d\times d}$ is a relatively small matrix. In contrast, in [51, 52], not only do their theoretical results rely on the fact that $m=O(T^7)$, but their algorithms also perform UCB/TS exploration on the entire network weight parameter, which is a $p=(L-1)m^2+md+m+d$ dimensional vector. This means [51, 52] need to compute the matrix inverse of a $p\times p$ matrix, which is impractical and certain approximations need to be done. In conclusion, by using shallow exploration, our method is much more computationally efficient than the neural contextual bandit algorithms in [51, 52]. We will add more discussion in the revision of our paper.
>
> ---
> **Q**: “Showing sublinear regret in the plot of Figure 1”
> **A**: The results on Statlog and MNIST (demonstrated in Figure 1a and Figure 1d) clearly show a sublinear regret. The sublinear regret on Magic and Covertype datasets are not clearly demonstrated due to the scale of the plot. To demonstrate this, we calculated the slope of the regret plot of Neural-LinUCB in the following table, which is strictly decreasing with time.
>
> Table: the slope of the regret plot of Neural-LinUCB at different time steps.
>
> |                   |  t=3000           |    t=6000        |       t=9000       |      t=12000     |      t=15000
> |    ---           |          ---           |         ---           |         ---            |        ---            |         ---
> |Magic         |0.22007333      | 0.21124083   |  0.20867667    |  0.20636042   |   0.20461091
> |Covertype  |0.35110667      |   0.32723083 |  0.31773667    |   0.31347167   |   0.31170485

---

> > ### Comment · Reviewer_7ZGH · 2021-08-25
> > **Dependence on $\lambda_0$? And more comments.**
> >
> > - I could not find the dependence on $\lambda_0$ in the regret bound. Can you state the exact dependence on $\lambda_0$, the lower bound on the minimum eigenvalue of the NTK matrix?
> > - Assumption 4.3 is not mild in the online setting as opposed to what the authors argue. The authors are not assuming any stochastic assumptions on contexts in the problem settings. Then, the question is "can Assumption 4.3 hold in adversarial settings?" What if my action set (context set) is fixed across time?
> > - There is still a structural assumption on the reward function while the authors argue that their method is for an "unknown" reward function. The reward function has to be an "inner product between a deep represented feature vector and an exploration weight parameter" hence, a linear function of "deep represented" feature vector. I believe that the method cannot guarantee a regret for an arbitrary reward function. The claims in the abstract and the introduction can be misleading to the readers.

---

> > > ### Author Response · Authors · 2021-08-26
> > > **Response to “Dependence on lambda_0 and other questions”**
> > >
> > > Thank you for raising these additional clarifying questions. We address them point by point in the following paragraphs.
> > >
> > > ---
> > > **Q**: “What is the dependence of the regret bound on $\lambda_0$?”
> > > **A**: The regret bound itself does not depend on $\lambda_0$. This quantity appears in the requirement of the lower bound of the width $m$. According to our proof, the dependence of $m$ on $\lambda_0$ should be $1/\lambda_0^4$. Currently, we omit it as a constant in the presentation of the theorem. We will make this dependence explicit in the main theorem in the revision.
> > >
> > > ---
> > > **Q**: “Assumption 4.3 is not mild in the online setting as opposed to what the authors argue. The authors are not assuming any stochastic assumptions on contexts in the problem settings. Then, the question is can Assumption 4.3 hold in adversarial settings? What if my action set (context set) is fixed across time?”
> > > **A**: First of all, we want to clarify that this assumption is standard in the analysis of neural network-based bandit algorithms, see the state-of-the-art results in [51, 52] for example. More importantly, this assumption is agnostic to the stochasticity of the context vectors (no matter whether the context vectors are given or arriving in an online fashion). In fact, this assumption holds if the neural tangent kernel is positive semi-definite (which is true because NTK kernel is indeed a Mercer kernel), and the data are non-degenerate (without two contextual vectors being identical or parallel). This has been proved in prior work (see e.g., [22, 53]).
> > >
> > > Thus, even if the contextual vectors are adversarial, which is exactly the case of contextual bandits studied in this paper, as long as the data are non-degenerate, Assumption 4.3 still holds. On the other hand, if the action set is fixed across time, then Assumption 4.3 does not hold. But in this case, we don’t need to define a $TK\times TK$ NTK matrix. Instead, we can average the rewards observed for the same contextual vector and use it as the response for the corresponding arm, and we only need to define a $K\times K$ NTK matrix instead of $TK\times TK$. As long as the $K$ contextual vectors in the fixed action set are non-degenerate, the $K\times K$ NTK matrix is still positive definite. In order to analyze the regret of this case, our proof needs to be modified, but the outline will be very similar.
> > >
> > > ---
> > > **Q**: “I believe that the method cannot guarantee a regret for an arbitrary reward function”
> > > **A**: You are correct that our method may not guarantee a regret for arbitrary reward functions. By “unknown” reward function, we did not mean arbitrary function.  As we have commented in our submission, our Neural-LinUCB works as long as the deep representation of the feature vector and the uncertainty weight parameter can be decoupled (see Line 273 in the paper). We will make it clear in the revision to show that we are dealing with unknown but structural reward functions.
> > >
> > > Please let us know if you have any other questions. Thanks.

---

> > > ### Author Response · Authors · 2021-08-31
> > > **Followup on the previous discussion**
> > >
> > > Thank you for all your comments and suggestions! We believe our response has addressed all your questions. Please let us know if you have any other questions, and we are happy to discuss more. If you’re satisfied with our response, we sincerely hope you could kindly re-consider the rating.

---

### Decision · Program_Chairs · 2021-09-27

**Decision:**

Reject

**Comment:**

This work presents a neural contextual bandit algorithm with computational improvements over prior work.  The work is very interesting, however reviewers, throughout both the feedback phase but also the extensive internal discussion, ended up very split; some find the overall approach very exciting and sound, others are concerned about the concrete improvement over prior work, which is powered by additional assumptions and thus a bit hard to measure.  Independent of the final decision, this is a valuable paper and I urge the authors in either their revisions or in re-submission to work to solidify the improvement over prior work, for instance achieving computational benefits without the extra assumptions, or relaxing the assumptions and/or making the analysis with them and the discussion and justification of the assumptions more compelling.